# Current Therapeutic Results and Treatment Options for Older Patients with Relapsed Acute Myeloid Leukemia

**DOI:** 10.3390/cancers11020224

**Published:** 2019-02-14

**Authors:** Felicetto Ferrara, Federica Lessi, Orsola Vitagliano, Erika Birkenghi, Giuseppe Rossi

**Affiliations:** 1Division of Hematology, Cardarelli Hospital, 80128 Napoli, Italy; orsola.vitagliano@gmail.com; 2Department of Medicine, Hematology and Clinical Immunology Unit, University of Padua, 35153 Padua, Italy; lessi.federica@gmail.com; 3Division of Hematology, Spedali Civili, 25123 Brescia, Italy; erika.borlenghi@gmail.com (E.B.); giuseppe.rossi@asst-spedalicivili.it (G.R.)

**Keywords:** acute myeloid leukemia, older patients, relapse, new drugs

## Abstract

Considerable progress has been made in the treatment of acute myeloid leukemia (AML). However, current therapeutic results are still unsatisfactory in untreated high-risk patients and poorer in those with primary refractory or relapsed disease. In older patients, reluctance by clinicians to treat unfit patients, higher AML cell resistance related to more frequent adverse karyotype and/or precedent myelodysplastic syndrome, and preferential involvement of chemorefractory early hemopoietic precursors in the pathogenesis of the disease further account for poor prognosis, with median survival lower than six months. A general agreement exists concerning the administration of aggressive salvage therapy in young adults followed by allogeneic stem cell transplantation; on the contrary, different therapeutic approaches varying in intensity, from conventional salvage chemotherapy based on intermediate–high-dose cytarabine to best supportive care, are currently considered in the relapsed, older AML patient population. Either patients’ characteristics or physicians’ attitudes count toward the process of clinical decision making. In addition, several new drugs with clinical activity described as “promising” in uncontrolled single-arm studies failed to improve long-term outcomes when tested in larger randomized clinical trials. Recently, new agents have been approved and are expected to consistently improve the clinical outcome for selected genomic subgroups, and research is in progress in other molecular settings. While relapsed AML remains a tremendous challenge to both patients and clinicians, knowledge of the molecular pathogenesis of the disease is fast in progress, potentially leading to personalized therapy in most patients.

## 1. Introduction

Acute myeloid leukemia (AML) occurs more commonly in elderly people. Median age at diagnosis is over 65 years, and incidence progressively increases with age, such that more than 40% of patients are currently diagnosed over the age of 70 [1,2,3]. Apart from accrual into experimental trials, therapeutic options in these patients include intensive chemotherapy (ICT) followed by allogeneic stem cell transplantation (SCT), hypomethylating agents (HMAs) such as azacytidine (AZA) and decitabine (DAC), low-dose cytarabine (LDARAC), and best supportive care (BSC), including hydroxyurea for the control of leukocytosis and transfusion support [4,5,6]. New treatment options, including venetoclax in combination with HMAs or LDARAC [7] and glasdegib in combination with LDARAC [8], have recently been approved by the Food and Drug Administration (FDA) for the treatment of newly diagnosed patients aged over 75 years and/or ineligible for intensive induction, where the treatments are expected to improve the clinical outcome. Following administration of conventional induction ICT, complete remission (CR) rates ranging from 40% to 70% are currently reported in untreated patients, depending on cytogenetics and molecular characteristics at diagnosis [9,10]. CR rate, including CR with incomplete hematological recovery (CRi), drops to 20%–25% after treatment with HMA and to less than 15% with LDARAC [11,12,13,14]. In spite of the lower CR rate, survival was quite similar in different studies comparing ICT to HMA, due to the possibility of achieving substantial clinical benefit with HMAs if hematological improvement and, to a lesser extent, stable disease are observed in the absence of CR [15,16,17]. Regardless of the initial treatment, recurrence of AML after CR still represents a major obstacle to overcome when cure is the objective of the treatment, independent of age and biomolecular characteristics at diagnosis [18,19,20,21]. However, age over 65 also represents a major adverse prognostic factor at the time of relapse. This may be explained by excessive treatment toxicity resulting in reluctance by clinicians to treat very elderly or unfit patients, higher AML resistance related to more frequent adverse karyotype and/or precedent myelodysplastic syndrome (MDS), and preferential involvement of chemorefractory early hemopoietic precursors in the pathogenesis of the disease [22]. Overall, the cumulative incidence of relapse approaches 60% at three years for patients in the European LeukemiaNet (ELN) favorable-risk category and exceeds 85% for those in the adverse-risk category [23,24,25]. In our experience, from a series of 80 patients (median age: 69, range: 65–84) treated with various approaches, the median survival from relapse was six months (Figure 1), with a small survival advantage for patients with core binding factor (CBF) AML, (i.e., AML with t[8;21] or inv[16]/t[16;16]) accounting for 8% of cases and AML relapsing with low marrow blast cell count, resembling myelodysplastic syndrome, as shown in Figure 2. While there is general agreement on the opportuneness of administering intensive salvage therapy, aimed to achieve second CR (CR2) followed by SCT to relapsed young-adult patients, the potential benefits—if any—deriving from the aggressive management of relapse of AML in elderly individuals are still unclear, and a remarkable uncertainty still impacts medical decisions. This results in different strategies related to patients’ and physicians’ attitudes and ranging in intensity, from conventional salvage chemotherapy based on intermediate–high-dose ARA-C to BSC [19]. Ideally, all poor-risk AML patients—particularly older ones with relapsed disease—would be enrolled in clinical trials based on the use of new agents. However, in daily practice, different factors represent common obstacles, including patient frailty and comorbidities, caregiver availability, and social support dynamics. In addition, protocol eligibility criteria are often stringent and account for further exclusion.

## 2. Intensive Salvage Chemotherapy and Hypomethylating Agents

Overall, ICT should be reserved for a small minority of patients not allografted for different reasons in CR1, in whom allogeneic transplant is feasible once CR2 has been achieved. There is increasing evidence supporting the utility of SCT in fit, older patients after CR achievement following intensive induction or HMA; accordingly, eligible patients should receive the procedure in CR1 with curative intent [26,27,28]. As a consequence, SCT in CR2 is unavoidably applicable in a negligible minority of cases. This further limits the role of salvage ICT, which should be reserved for patients with ELN-favorable criteria and CR1 lasting for more than one year, or for the very small number of patients where allogeneic SCT had been planned as part of their treatment program at diagnosis and who have not experienced prohibitive toxicity during first-line therapy.

The role of HMAs is well-established in the frontline treatment of older patients with AML, including bridging to transplant [29], while no randomized trials have been performed in refractory or relapsed disease. In the absence of prospective studies, different retrospective observations have demonstrated the potential utility of either AZA or DAC. Most relevant data derive from the analysis of a large international multicenter retrospective database, focusing on the effectiveness of HMA as well as on predictors of response and overall survival (OS). A total of 655 patients, including 290 refractory and 365 relapsed patients, were given AZA (57%) or DAC (43%). Median age at diagnosis was 65 years. The CR rate (CR + CRi) was 16%, while hematologic improvement was observed in 8.5%. Median OS was 6.7 months and strictly related to best response achieved (25.3 months for patients achieving CR and 14.6 months for CRi). The presence of more than 5% of blasts in peripheral blood and >20% blasts in the bone marrow were significantly associated with shorter OS in the multivariate analysis, while a 10-day schedule of DAC induced a higher response rate [30].

Similarly, we retrospectively reviewed clinical records of 79 patients treated with HMAs as salvage therapy at nine institutions in Italy, with a median age of 64 and with secondary AML in 29%. According to ELN criteria, 10 patients were favorable-risk, 35 were intermediate-, and 30 were adverse-risk. All patients had been given ICT at the onset of disease; 61% of patients received HMA as second-line therapy, 26% as third-line, and 13% were beyond the third line. Note that 18% of patients had received SCT before HMA. Overall response rate (CR + CRi) was 18%. Median OS in patients with relapsed disease was 14.9 months vs. 5.1 for refractory ones (Figure 3). Best results were observed in 46% of patients, who showed either CR, CRi, hematologic improvement, or stable disease after salvage HMA [31]. These data seem to compare favorably with intensive salvage chemotherapy, and suggest that HMAs represent an acceptable therapeutic option for the selected population of relapsed elderly patients, especially those previously treated with ICT and not suitable for allogenic bone marrow transplantation. Finally, AZA has been proven as potentially useful for relapse after SCT in either AML or MDS, with a response rate and survival which do not substantially differ from those reported after ICT and would represent the preferred option in the adverse subset of elderly patients with relapsed disease after SCT [32].

An emerging clinical challenge concerns the treatment of older AML patients who relapse after CR or progress after any response following initial therapy with HMAs. For this category, we should consider that ICT in most cases was already excluded at the time of diagnosis and therefore it should be even more so at the time of relapse. Furthermore, results of ICT in patients treated with AZA for MDS and who progressed to AML while on therapy are disappointing, with low CR rate and high treatment-related mortality [33,34,35]. In the absence of a clinical trial based on the use of experimental drugs, BSC and/or hydroxyurea for the control of leukocytosis still represent the best option for this subset of patients, with the aim of improving quality of life in an outpatient setting. Recently, evidence has been provided for the use of venetoclax in combination with HMAs in patients with refractory/relapsed AML treated outside of a clinical trial. In a small series of 33 consecutive adults with a median age of 62 years (range 19–81), in which 20 out of 33 (61%) have been pretreated with HMAs, CR + CRi accounted for 51% with a median survival of six months [36].

## 3. New Approaches for the Treatment of Relapse of AML in Older Patients

### 3.1. FLT3 Inhibitors

In the past decade, there has been considerable progress in understanding the molecular pathogenesis of AML, which has led to the development of potential therapeutic targets so that selective treatment approaches aimed at rational and personalized treatment strategies are now available [37,38,39,40]. In the past two years, different new agents for AML have become available for newly diagnosed or relapsing/refractory patients, and others are the object of clinical investigation. The last approval by the FDA for refractory/relapsed AML patients refers to gilteritinib (G), a powerful FLT3 inhibitor, for the treatment of adult patients with relapsed or refractory AML with an FLT3 mutation as detected by an FDA-approved test. The incidence of FLT3/ITD mutations varies according to age and clinical risk group, being less common in pediatric AML and in AML arising from an antecedent myelodysplastic syndrome. However, the frequency of the mutation in the elderly accounts for more than 20%, so a non-negligible percentage of patients would benefit from FLT3 inhibitors [41,42,43]. Approval of G was based on an interim analysis of the ADMIRAL trial (NCT02421939), which included 138 adult patients with relapsed or refractory AML carrying an FLT3–ITD, D835, or I836 mutation. G was given orally at a dose of 120 mg daily until unacceptable toxicity or a lack of clinical benefit was observed. After a median follow-up of 4.6 months (range: 2.8 to 15.8), 29 patients (21%) achieved CR or CRi and 33 (31.1%) sustained red blood cell and transfusion independence. Toxicity was acceptable; the most common adverse effects (>20% of patients) consisted of myalgia/arthralgia, transaminase increase, fatigue/malaise, fever, noninfectious diarrhea, dyspnea, edema, rash, pneumonia, nausea, stomatitis, cough, headache, hypotension, dizziness, and vomiting [44]. Note that approval was based on a phase 2 non-randomized study, because available treatment options are limited and largely unsatisfactory for patients with relapsed/refractory FLT3–ITD AML. These data, along with the oral formulation, suggest the possibility of managing selected patients on outpatient basis and make treatment with G particularly attractive in the older AML population.

More recently, the FDA granted a priority review designation to quizartinib (Q), a new FLT3 inhibitor for the treatment of adult patients with relapsed/refractory *FLT3* ITD positive AML [45]. The efficacy and safety of single-agent Q were evaluated in the phase 3 Quantum R randomized trial, aimed at comparison of Q vs. investigator choice (IC), including conventional salvage ICT or LDARAC. In 367 patients randomized with a 2:1 ratio (245 to Q and 122 to the control arm), the median OS was 6.2 months, with an estimated 12-month OS probability of 27% vs. 20% in Q and IC arms, respectively; median event free survival (EFS) was 6.0 vs. 3.7 (95% CI, 0.4–5.9) weeks, respectively. The superiority of Q was confirmed by analyses across subgroups, including FLT3 allelic ratio, prior HSCT, AML risk score, and response to prior therapy. The CR + CRi rate was 48% in Q and 27% in the IC arms (nominal *p* = 0.0001) and the transplant rate was 32% and 12% in Q and SC arms, respectively. Toxicity was comparable between the two arms and only two patients discontinued Q due to QTcF prolongation [46]. These data strongly suggest that Q may represent an important therapeutic option for older patients with refractory/relapsed AML in the near future.

### 3.2. IDH1 Inhibitors

Recurring mutations in isocitrate dehydrogenase (IDH) genes are detected in approximately 20% of adult patients with AML and 5% of adults with MDS [47,48]. The prognostic significance of mutant IDH is controversial, but appears to be influenced by co-mutational status and the specific location of the mutation [49,50]. For relapsing AML patients harboring a mutation in IDH 1 or 2 (IDH1/2), potential treatment options have undergone a paradigm shift away from intensive cytotoxic chemotherapy to targeted therapy with selective inhibitors, such as enasidenib (ENA) for IDH2 or ivosidenib (IVO) for IDH1, both recently approved by FDA [51,52,53]. In addition, the possibility of combining aggressive or attenuated chemotherapy with either ENA or IVO is currently the object of investigation in ongoing clinical trials. ENA was approved by the FDA for relapsed or refractory AML with an IDH2 mutation, concurrently with a companion diagnostic, the RealTime IDH2 Assay, used to detect the IDH2 mutation. Approval was based on Study AG221-C-001, an open-label, single-arm, multicenter clinical trial that accrued 199 adults with relapsed or refractory AML. Patients received ENA orally at 100 mg/day. Twenty-three percent of patients achieved CR or CRi lasting a median of 8.2 months, with 19% of patients having a CR lasting a median 8.2 months, and 4% with a CRi lasting a median 9.6 months. Noticeably, among 157 patients who were transfusion-dependent at the beginning of the trial, 34% no longer required transfusions during at least one 56-day time period on treatment. The most common adverse reactions occurring in more than 20% of patients were gastrointestinal and included nausea, vomiting, diarrhea, elevated bilirubin, and decreased appetite [54].

A recent phase 1 dose-escalation clinical trial with IVO has prompted approval by FDA for the treatment of patients with *IDH1*-mutated AML in the relapsed and refractory setting due to favorable results [55]. In the refractory/relapsed population (179 patients), the rate of CR was 21.8% and CRi 11.7%. With a median follow-up of 14.8 months, the median OS in the primary efficacy population was 8.8 months; the 18-month survival rate was 50.1% among patients who had CR or CRi. Estimates of median OS were 9.3 months among patients obtaining CR and 3.9 months among patients who did not have a response. Transfusion independence was attained in 29 of 84 patients (35%). Among 34 patients who had a complete remission or complete remission with partial hematologic recovery, 7 (21%) had no residual detectable IDH1 mutations on digital polymerase-chain-reaction assay. No pre-existing co-occurring single gene mutation predicted clinical response or resistance to treatment. Treatment-related adverse events of grade 3 or higher that occurred in at least three patients included QT interval prolongation in 7.8% of the patients, the IDH differentiation syndrome in 3.9%, anemia (2.2%), thrombocytopenia or a decrease in the platelet count (3.4%), and leukocytosis (1.7%). These results suggest that in patients with advanced IDH1-mutated relapsed or refractory AML, IVO at a dose of 500 mg daily was associated with sustained clinical benefit, including transfusion independence, durable remissions, and molecular remissions in some patients with CR. Note that these results compare favorably with those described for salvage intensive chemotherapy, resulting in significantly lower response rate and survival [56]. The incidence of differentiation syndrome (DS) with IVO and ENA in the treatment of patients with relapsed or refractory AML has recently been evaluated through a systematic analysis by the FDA [57]. Criteria previously established for acute promyelocytic leukemia (APL) [58] were utilized so that patients with two or three criteria were classified as having moderate DS and patients with at least four criteria were classified as having severe DS. DS was excluded in cases with an alternative explanation (e.g., septic shock). Overall, 72/179 (40%) cases of potential DS for IVO and 86/214 (40%) for ENA were identified by FDA reviewers [56]. This contrasts with the DS incidence of 11% (19/179) for IVO (52) and 12% (26/214) for ENA reported by investigators and review committee determination, respectively [59]. Of note, for both IVO and ENA, the CR + CRi rate in patients with DS was numerically lower than that in patients without DS (IVO: 18%; ENA: 18%), while age, demographics, and cytogenetic risk of patients with FDA-identified DS were similar to those of patients without DS. As in APL, leukocytosis was not always present. Obviously, earlier and more careful recognition of signs and symptoms of DS may lead to earlier diagnosis and treatment, which may decrease severe complications and mortality.

NPM1 mutation confers a relatively more favorable prognosis in patients with FLT3-unmutated AML. Its frequency is higher in younger patients, but even after the age of 75, NPM1-mutated AML may account for more than 30% of cases [60,61]. The long-term efficacy of iCT in these patients is debated, and may be related to the mutational status of leukemia, owing to the frequent co-occurrence of epigenetic mutations, most frequently in the DNMT3α gene. Regimens other than iCT with limited toxicity like actinomycin D or all-trans retinoic acid have proven effective in occasional patients with relapsed or refractory disease, or considered unfit for iCT [62,63]. While prospective trials are underway to confirm these results, these agents may be considered as an alternative to BSC in patients with advanced NPM1-mutated AML. Molecular targets and therapeutic results of recently approved new agents for refractory/relapsed AML are summarized in Table 1.

## 4. Conclusions

Following relapse, the prognosis of patients with AML remains extremely poor and curative options are limited, especially in the older patient population. A minority of patients over 65 years are actually eligible for SCT in CR1 and significantly less at relapse, therefore new strategies are needed in order to improve therapeutic results. More than 15 years ago, we demonstrated that intensive salvage therapy was not indicated in the majority of relapsed older patients with AML, namely in those with CR1 duration less than one year and/or with unfavorable cytogenetics [64]. As shown in Figure 4, in addition to ICT just for the very fit patient candidate for SCT and non-intensive treatment with hypomethylating agents, for patients with FLT3 and IDH1/2 mutations we now have the possibility of offering new agents which have been found to be more effective and probably less toxic than conventional salvage CT. In addition, the oral formulation represents a substantial advantage for the older AML population because of the possibility of managing a substantial number of patients in an outpatient setting. Finally, unlike ICT, clinical benefit can be achieved with these agents also in the absence of CR, with reduction of or independence from transfusion support and improved quality of life. It should be considered that the above genetic patterns account for no more than 40% of the whole older AML patient population and results need to be confirmed on larger patient series and in real-life studies. However, the landscape of AML treatment is undergoing dramatic evolution due to progress in understanding the molecular pathogenesis of the disease and the introduction of additional new agents either at diagnosis or relapse. In this regard, high-risk patients—especially older patients with refractory or relapsed disease—represent an ideal field of clinical investigation.

## Figures and Tables

**Figure 1 cancers-11-00224-f001:**
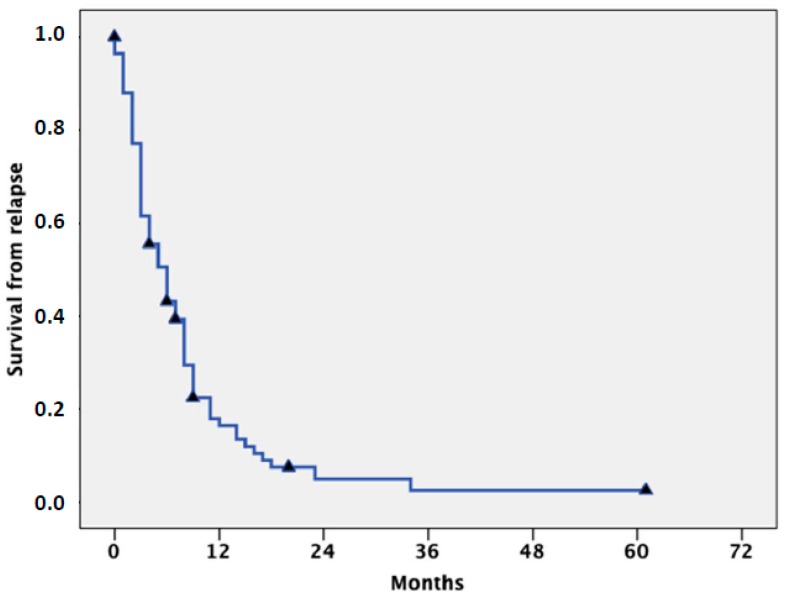
Survival from relapse of 80 consecutive relapsed acute myeloid leukemia (AML) elderly patients, previously treated with intensive chemotherapy (median age: 69 years, range: 65–84).

**Figure 2 cancers-11-00224-f002:**
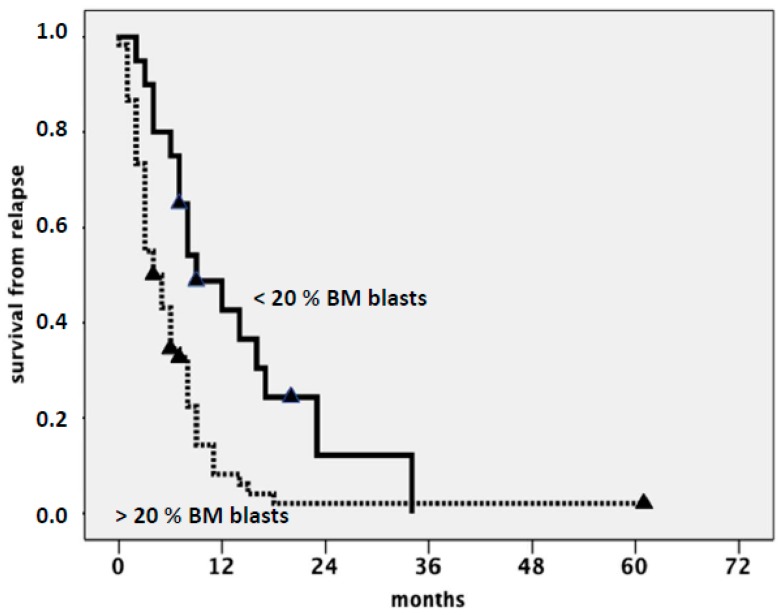
Survival from relapse according to bone marrow blast count at relapse (> or ⊇20%).

**Figure 3 cancers-11-00224-f003:**
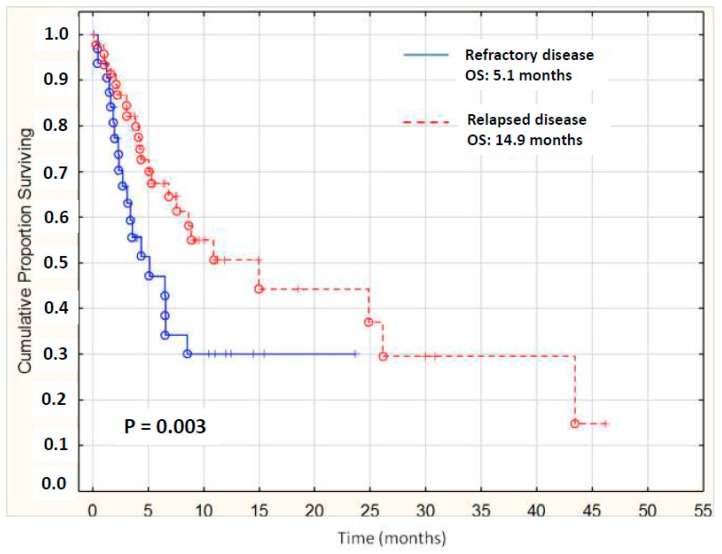
Survival from relapse of 79 AML elderly patients treated with hypomethylating agents (HMAs); median of relapsed patients was significantly higher than refractory ones (14.9 vs. 5.1 months).

**Figure 4 cancers-11-00224-f004:**
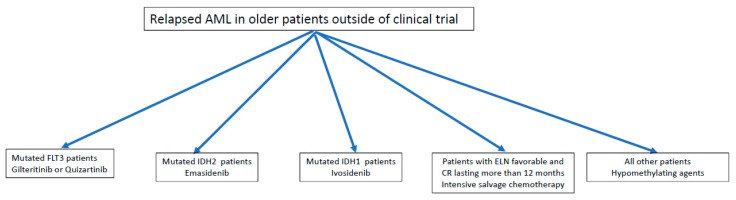
Potential therapeutic algorithm for older patients with relapsed AML elderly. ELN: European LeukemiaNet.

**Table 1 cancers-11-00224-t001:** Molecular targets and results of recently approved new drugs for the treatment of relapsed elderly patients with AML. AZA: azacytidine; CR: complete remission; CRi: CR with incomplete hematological recovery; DAC: decitabine.; ORR: overall response rate; HCT: hemopoietic cell transplantation.

Drug [Ref.]	Target	ORR (CR + CRi)	Median Survival (Months)	Other Benefits
Gilteritinib [43]	FLT3	21%	4.6	Sustained transfusion independence (31%)
Quizartinib [45]	FLT3	48%	6.2	Benefit across subgroups, including varying allelic ratio, prior HCT, AML risk score, and response to prior therapy
Ivosidenib [54]	IDH1	33%	8.8	Sustained transfusion independence (35%)
Enasidenib [53]	IDH2	23%	8.2	Sustained transfusion independence (34%)
AZA or DAC [30]	DNA methylation	16%	6.7	Hematological improvement (8%)
AZA/DAC + Venetoclax [36]	DNA methylation BCL2 regulation	51%	6.5	10% morphological leukemia free state

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
