# Peer review of "Current Therapeutic Results and Treatment Options for Older Patients with Relapsed Acute Myeloid Leukemia"

_cancers, 2019, doi:10.3390/cancers11020224_

Round 1
Reviewer 1 Report
In the review «Current therapeutic results and treatment options for older patients with relapsed acute myeloid leukemia» the authors raise the problems encountered when treating elderly and/or unfit patients with AML. These patients have a high risk of relapse, treatment-related mortality, age-related co-morbidity and, in general, short survival after diagnosis. In the manuscript the authors discus the newest treatment options, especially related to Flt3 and IDH1/2 inhibitors.
This is a comprehensive review on the topic, and I have only minor comments.
General comments
In section 2 (lines 80-89) the authors mention the possibility of allogeneic transplantation after second remission. Since the authors also state that eligible patients receive this treatment after CR1, I was wandering if they could say more about the reasons why clinicians do not transplant all eligible patients after CR1 but wait to CR2 for some patients (the risks are not exactly decreasing).
For presentation of results in sections 2 and 3, summarizing tables would help the readers to extract the most interesting numbers.
The authors focus on the promising Flt3-inhibitors gilteritinib and quizartinib, and the IDH1/2-inhibitors enasidenib and ivosedenib. Another drug, which currently is much discussed, is the Bcl-2-inhibitor venetoclax. Could this be included, or do the authors consider it to be out of the scope? It is mentioned in the introduction.
Minor comments
The meaning of the sentence (lines 18-21) in the abstract is difficult to grasp. Please revise.
Line 38: consider “newly diagnosed” instead of “naïve” patients.
Line 41: what to the authors mean by “untreated patients”? Can it be “patients with de novo AML”?
Line 59: the word “patients” or “cases” appears to be missing after “8% AML”.
Figure 2: The dashed line in the figure appears to correspond to the line in figure 1. Can the authors mention this in the figure caption?
Section 3: consider breaking the text up in sections with subheadings as different types of specific inhibitors are discussed.
Figure 4: “Mutated IDH1 patients” appears twice in the figure.
Author Response
Reviewer 1
We have specified that allo-SCT should be performed when feasible in CR1 and is minimally applicable in CR2.
A summarizing table has been added as sections 2 & 3 are concerned
The combination VEN + HMAs had been excluded from analysis in that best results have been reported and the combination is now available in newly diagnosed AML patients, for whom FDA and EMA registration has been recently achieved. In relapsed/refractory patients data are limited; however, the possibility of using this combination has been considered and added with references.
The sentence on line 18-21 has been modified
The sentence on line 38 has been modified
NaĂŻve has been changed
Section 3 has been modified as suggested
Fig 4 has been corrected
Reviewer 2 Report
The authors provide a timely and thorough summary of the data surrounding the recently FDA-approved, molecularly-targeted drugs for specific molecular subgroups for AML. There are a few issues that the authors may wish to address:
The authors propose an algorithm for incorporating these agents into a "standard of care" approach for older patients with relapsed AML. However, any implication that we are ready for these agents to be used in a routine fashion outside of the clinical trials arena is very premature and must be toned down significantly. These agents are now being used widely in clinical practice on the basis of very little data and modest clinical results. Indeed, these agents should not be considered standards of care in any format and it is not appropriate for these agents to be used widely and routinely without data capture and continuing evaluation. The tone of this review is far too positive, and both the last sentence of the Introduction (which seems to excuse the broad use of all of these agents at this time) and Figure 4 (which proposes an algorithm for such broad use) need to be either deleted or restructured to emphasize the prematurity of accepting the knowledge about these drugs as being complete and "ready for prime time."
A table summarizing the data for each agent -- e.g., molecular target, response as single agent and/or in combination -- would enhance the utility of this review article.
Author Response
Reviewer 2
A summarizing table has been added as sections 2 & 3 are concerned.
In general we agree with reviewer 2 that “these agents should not be considered standards of care in any format and it is not appropriate for these agents to be used widely and routinely without data capture and continuing evaluation”. However, results with conventional current salvage therapy are extremely poor in relapsed/refractory AML of the elderly; in addition in the conclusions we have specified that “results need to be confirmed on larger patient series as well as in real life studies”.
Round 2
Reviewer 2 Report
The authors have addressed the issues raised by the reviewers in a thorough fashion.